# Eckol Alleviates Intestinal Dysfunction during Suckling-to-Weaning Transition via Modulation of PDX1 and HBEGF

**DOI:** 10.3390/ijms21134755

**Published:** 2020-07-03

**Authors:** Sang In Lee, In Ho Kim

**Affiliations:** 1Department of Animal Biotechnology, Kyungpook National University, Gyeongsangbuk-do, Sangju 37224, Korea; silee78@knu.ac.kr; 2Department of Animal Resource and Science, Dankook University, Chungcheongnam-do, Cheonan-si 330-714, Korea

**Keywords:** barrier function, eckol, HBEGF, ethanol extract of dried *E. cava*

## Abstract

Maintaining intestinal health in livestock is critical during the weaning period. The precise mechanisms of intestinal dysfunction during this period are not fully understood, although these can be alleviated by phlorotannins, including eckol. This question was addressed by evaluating the changes in gene expression and intestinal function after eckol treatment during suckling-to-weaning transition. The biological roles of differentially expressed genes (DEGs) in intestinal development were investigated by assessing intestinal wound healing and barrier functions, as well as the associated signaling pathways and oxidative stress levels. We identified 890 DEGs in the intestine, whose expression was altered by eckol treatment, including pancreatic and duodenal homeobox (PDX)1, which directly regulate heparin-binding epidermal growth factor-like growth factor (HBEGF) expression in order to preserve intestinal barrier functions and promote wound healing through phosphoinositide 3-kinase (PI3K)/AKT and P38 signaling. Additionally, eckol alleviated H_2_O_2_-induced oxidative stress through PI3K/AKT, P38, and 5’-AMP-activated protein kinase (AMPK) signaling, improved growth, and reduced oxidative stress and intestinal permeability in pigs during the weaning period. Eckol modulates intestinal barrier functions, wound healing, and oxidative stress through PDX/HBEGF, and improves growth during the suckling-to-weaning transition. These findings suggest that eckol can be used as a feed supplement in order to preserve the intestinal functions in pigs and other livestock during this process.

## 1. Introduction

The suckling-to-weaning transition contributes to intestinal dysfunction in livestock, which undermines animal health, growth, and feed intake [1]. During this transition, the intestine undergoes marked physiologic changes in structure and function, including villous atrophy and crypt elongation, which decrease its absorptive capacity, thereby influencing feeding efficiency [2,3,4]. Alleviating intestinal dysfunction during this process is important, given the direct relationship between animal health and economic productivity.

The intestinal epithelium is composed of single-layered columnar epithelial cells that are sealed by junctional complexes, including tight and adherens junctions, in close proximity to the apical and lateral sides of the paracellular space [5]. It functions as a barrier against harmful substances, including pathogenic bacteria and food allergens [6]. Intestinal dysfunction that is induced by the suckling-to-weaning transition can disrupt these junctional complexes, thereby allowing for the passage of macromolecules and pathogens through the epithelium into the body, which can influence animal growth and health status [7,8]. Preserving intestinal health minimizes the adverse effects of weaning-induced stress. To this end, nutritional strategies, such as optimizing dietary proteins or energy content and the use of feed additives, have been implemented [9,10,11]. Antibiotics, as feed additives, are broadly used in the pig industry to enhance intestinal health that is challenged by weaning-induced stress. Many alternatives, including probiotics, organic acids, and polyphenols, are used as substitutes due to a ban on antibiotic use in feedstuffs, although in-feed antibiotics are used to reduce weaning-induced stress and enhance growth performance [12,13,14]. Among these candidates, polyphenols may have the greatest potential to enhance gut intestinal health [15].

Polyphenols have been investigated for their potential in promoting gut health and regulating the intestinal nutrient absorption, as well as lipid and bone metabolism [16]. Phlorotannins are oligomeric polyphenols that consist of phloroglucinol units that are found in brown algae (*Ecklonia* sp.), including *Eckolina cava*; this is an edible marine brown alga species found in the ocean off the coasts of Japan and Korea. These compounds have anti-diabetic, anti-cancer, antioxidant, antibacterial, and radio-protective properties, as well as inhibitory effects against HIV [17,18,19,20]. A previous report showed that compounds isolated from the ethanol extract of dried *E. cava* (EEEC), including phloroglucinol, eckol, phlorofucofuroeckol, and dieckol, exhibit strong antiviral activity against porcine epidemic diarrhea virus by inhibiting viral entry and/or replication, and suppressing adipogenesis by downregulating C/EBPα in adipocytes [21,22]. In addition, a previous report suggested that *E. cava* has beneficial effects on growth performance, cecal microflora, and intestinal morphology in weaning pigs [23]. However, to our knowledge, there is no study investigating the effects of these compounds on the intestinal function of piglets during the suckling-to-weaning transition, as this is the most active period of small intestinal epithelium development.

To this end, we evaluated the effects of EEEC on the small intestine during the said transition period by gene expression profiling. Differentially expressed genes (DEGs) upon EEEC treatment were further analyzed in order to determine their roles in small intestine development.

## 2. Results

### 2.1. Identification and Validation of DEGs

We used high-throughput sequencing to identify DEGs that were expressed in response to EEEC by comparing the gene expression profiles of small intestinal tissue with or without EEEC treatment for 14 d. Of the 890 annotated DEGs, 639 were upregulated and 251 were downregulated (Figure 1A).

A gene ontology (GO) enrichment analysis showed that DEGs were related to calcium, heme, iron, heparin, carbohydrate, and lipid binding, as well as chemokine, transporter, serine-type endopeptidase inhibitor, and hormone activities (Appendix A). Genes within the “cellular component” category were mainly associated with the “extracellular region”, “integral component of plasma membrane”, “cell surface”, and “cell” terms (Appendix A), whereas the “biological process” terms included “inflammatory response”, “immune response”, “innate immune response”, “oxidation-reduction process”, “cell surface receptor signaling pathway”, “cell adhesion”, “cytokine-mediated pathway”, “response to lipopolysaccharide”, and “chemokine-mediated signaling pathway” (Appendix A). The Kyoto Encyclopedia of Genes and Genomes pathway analysis showed that the proteins were mainly associated with the following categories: cytokine–cytokine receptor, neuroactive ligand–receptor, and extracellular matrix–receptor interactions; peroxisome proliferator–activated receptor γ, chemokine, Janus kinase-signal transducer and activator of transcription (STAT), tumor necrosis factor, Toll-like receptor, and transforming growth factor (TGF)-β signaling; inflammatory bowel disease (IBD); and, retinol metabolism (Appendix A).

We verified the expression of the top 10 DEGs in the small intestine with or without EEEC treatment through qRT-PCR (Figure 1D) and confirmed that *lectin, galactoside-binding, soluble 13* (*p* < 0.01), *pheromaxein C subunit* (*p* < 0.01), *PDX1* (*p* < 0.01), *solute carrier family 22 member A7* (*p* < 0.01), *adenylate cyclase activating polypeptide 1* (*p* < 0.01), *resistin* (*p* < 0.05), heparin-binding epidermal growth factor-like growth factor (*HBEGF)* (*p* < 0.05), *RNA-binding protein 7* (*p* < 0.06), *Spalt-like transcription factor 1* (*p* < 0.05), and *A disintegrin and metalloproteinase with thrombospondin motifs 20* (*p* < 0.05) were more strongly upregulated in the EEEC-treated group as compared with the untreated group (Figure 1D).

### 2.2. Eckol Induces PDX1 and HBEGF Expression

We next evaluated the toxicity of EEEC, eckol, dieckol, and phlorofucofuroeckol in IPEC-J2 cells while using a cell viability assay. Pretreatment for 24 h with 50 µM of EEEC, 100 µM of eckol and dieckol, or 200 µM of phlorofucofuroeckol reduced cell viability (Figure 2A). Based on these results, 10 µM of EEEC, 50 µM of eckol and dieckol, and 100 µM of phlorofucofuroeckol were selected as the safe dosages for subsequent experiments. *HBEGF* expression was induced by treatment with 10 µM of EEEC (*p* < 0.01) and 50 µM of eckol (*p* < 0.05) (Figure 2B). These results indicate that eckol is the main component of EEEC that mediates differential gene expression.

Indeed, eckol incubation with different concentrations (10 µM, 20 µM, 50 µM, 100 µM, and 200 µM) for 24 h resulted in a concentration-dependent increase in *PDX1* (Figure 3A,B) and *HBEGF* (Figure 3C,D) mRNA and protein levels.

### 2.3. Eckol-Mediated Induction of PDX1 Regulates HBEGF Expression

We cloned different lengths of porcine *HBEGF* gene promoter sequences into the firefly luciferase plasmid to evaluate the transcriptional activity of PDX1 in response to eckol treatment in order to investigate whether PDX1 activation directly regulates HBEGF expression in intestinal epithelial cells in response to eckol treatment. Luciferase activity was stimulated in the presence of the −2000, −1500, and −1000 sequences relative to that of the control (Figure 4A). Three regions (−954, −790, and −767) of the PDX1 binding sequence (TAAT) were identified between −1000 and −500 (Figure 4B). The deletion of the PDX1 binding sequence upstream of −790 reduced luciferase activity as compared with the control levels (Figure 4C), suggesting that it is essential for the *HBEGF* promoter’s basal transcriptional activity.

Next, we examined whether *PDX1* knockdown affects *HBEGF* expression. Three different siRNA sequences (siRNA-1, siRNA-2, and siRNA-3 with knockdown efficiencies of 13.32% ± 14.96%, 78.47% ± 16.99% (*p* < 0.05), and 5.35% ± 23.06%, respectively) against porcine *PDX1* reduced *PDX1* expression in IPEC-J2 cells more efficiently than a non-specific siRNA with no homology to porcine sequences (Figure 4D). According to these results, we used PDX1-siRNA-2 for subsequent experiments. After eckol treatment, *PDX1* knockdown reduced *HBEGF* expression (Figure 4E). Taken together, our results indicate that PDX1 directly regulates *HBEGF* expression by binding to the promoter region upstream of −790.

### 2.4. Eckol Enhances Intestinal Barrier Function and Promotes Wound Healing

We examined phosphoinositide 3-kinase (PI3K)/AKT activation (Figure 5A) and P38 (Figure 5B) signaling pathways to investigate the mechanism by which eckol modulates the expression and function of HBEGF in the intestine. Treating IPEC-J2 cells with eckol and HBEGF increased AKT phosphorylation, whereas treatment with the PI3K inhibitor LY294002 decreased AKT phosphorylation after eckol and HBEGF treatment, as compared with that in the untreated controls. Eckol and HBEGF treatment also increased P38 phosphorylation, but it was reversed after treatment with the P38 mitogen-activated protein kinase (MAPK) inhibitor SB202190.

Next, we examined the effects of eckol and HBEGF on intestinal barrier function by evaluating TEER, permeability, and tight junction protein expression in IPEC-J2 cells. LPS treatment for 1 h decreased TEER to less than that in the untreated control cells, an effect that was reversed after eckol and HBEGF treatment (Figure 5C). In contrast, the application of LY294002 and SB202190 reduced TEER in LPS-treated cells. LPS exposure for 1 h also reduced the permeability to FD-4, whereas it was reversed by eckol and HBEGF treatment (Figure 5D) and intensified by applying LY294002 and SB202190. Immunocytochemical and Western blot analyses showed that ZO-1 expression was downregulated relative to the control levels in IPEC-J2 cells that were treated with LPS for 1 h, but it was upregulated by eckol and HBEGF treatment and abrogated by LY294002 and SB202190 treatment (Figure 5E,F).

Intestinal wound healing depends on a precise balance among migration, proliferation, and differentiation of epithelial cells adjacent to the wound. Here, we investigated the effects of eckol and HBEGF on intestinal wound healing by assessing cell proliferation and migration while using IPEC-J2 cells. LPS treatment reduced cell growth, whereas adding eckol and HBEGF to the culture media increased cell proliferation (Figure 6A). Similarly, eckol and HBEGF treatment enhanced the migratory capacity of LPS-challenged cells, an effect that was diminished by applying LY294002 and SB202190 (Figure 6B).

We analyzed the expression of wound healing-related genes, such as *matrix metalloproteinase* (*MMP*) *2*, *MMP9*, and *Rho family GTPases* (*RND*) *3* in IPEC-J2 cells with or without eckol and HBEGF treatment to determine the effects of eckol and HBEGF on intestinal epithelial cell differentiation (Figure 6C). The *MMP2*, *MMP9*, and *RND3* levels were downregulated by LPS treatment relative to that in the untreated control cells, but they were increased by eckol and HBEGF treatment.

### 2.5. Eckol Protects against Oxidative Stress in the Intestine

We analyzed *heme oxygenase* (*HO*)*-1* and *manganese superoxide dismutase* (*MnSOD*) expression, as well as reactive oxygen species (ROS) production, in IPEC-J2 cells to investigate the effects of eckol on intestinal oxidative stress. H_2_O_2_ treatment for 1 h reduced the *HO-1* levels relative to that in untreated cells, but this was reversed by eckol treatment, and LY294002 had a similar effect (Figure 7A). The H_2_O_2_-treated cells showed higher DCF fluorescence as compared with that in the untreated control, and SB202190 treatment also increased DCF fluorescence; however, this was reversed by eckol treatment (Figure 7B). Thus, eckol alleviates H_2_O_2_-induced oxidative stress through PI3K/AKT signaling.

*MnSOD* expression was decreased, whereas DCF fluorescence was increased in H_2_O_2_-treated IPEC-J2 cells relative to control cell levels. Eckol treatment restored *MnSOD* levels and reduced DCF fluorescence, whereas 5’-AMP-activated protein kinase (AMPK) inhibitor treatment reversed these effects (Figure 7C,D). These results suggest that eckol alleviates H_2_O_2_-induced oxidative stress through AMPK signaling.

### 2.6. EEEC Supplementation Improves Intestinal Function in Pigs during Weaning

EEEC was administered as a dietary supplement to pigs in the weaning period, and we evaluated their growth, as well as the serum levels of stress markers, in order to investigate the in vivo effects of EEEC. Average daily gain (ADG) was higher in pigs fed with 0.05% and 0.1% EEEC than that in pigs fed with the control diet during phase I (days 1–7) and phase II (days 8–21) (*p* < 0.05; Figure 8A). At the end of the feeding trial, (day 42), the cortisol levels were lower in pigs that were fed with 0.05% (*p* < 0.05) and 0.1% (*p* < 0.05) EEEC, and epinephrine and norepinephrine levels were reduced in pigs fed with 0.1% EEEC (*p* < 0.05; Figure 8B). Pigs fed with 0.05% and 0.1% EEEC also had lower serum superoxide dismutase (SOD) and glutathione peroxidase (GPx) levels and higher serum malondialdehyde (MDA) levels on days 7 and 21, as compared with control animals (*p* < 0.05; Figure 8C). Thus, EEEC supplementation improves growth performance and reduces stress in pigs during the weaning period.

## 3. Discussion

The suckling-to-weaning transition is associated with various sources of stress, such as maternal separation, mixing, transportation/relocation, and dietary changes. These stressors collectively contribute to intestinal dysfunction, including intestinal villus atrophy, crypt hyperplasia, and increased intestinal permeability, which results in reduced feed intake and retarded growth, as well as increased disease susceptibility. Therefore, this period represents a major bottleneck in livestock production [1,3]. To minimize intestinal dysfunction during this transition, nutritional strategies, such as manipulating the dietary protein and non-starch polysaccharide contents, as well as adding antibiotics, zinc, probiotics, prebiotics, and polyphenols to livestock diet, have been applied to pig production [15,24]. Phlorotannins, such as eckol, dieckol, and phlorofucofuroeckol found in brown algae, have protective effects against oxidative stress-induced mitochondrial dysfunction and radiation-induced intestinal injury [25,26]. In the present study, we investigated the mechanism of action of EEEC in intestinal development during the suckling-to-weaning transition.

Our transcriptome analysis identified 639 upregulated and 251 downregulated genes in response to EEEC treatment. These DEGs were enriched in various GO categories, and the expression patterns of the top 10 genes were confirmed in the small intestine. Consistent with our results, previous studies demonstrated that polyphenols alter the expression of genes that are related to “Wnt signaling”, “chemokine activity”, “cytokine activity”, “bone morphogenetic protein”, “TGF-β signaling”, “apoptosis”, “adipogenesis”, “cytokine response”, “inflammatory response”, and “proliferation” [27,28]. We also analyzed *PDX1* and *HBEGF* expression in intestinal epithelial cells treated with each phlorotannin to investigate the role of individual EEEC components. Based on the results, eckol-induced *PDX1* and *HBEGF* expression to a greater extent than dieckol and phlorofucofuroeckol; hence, eckol was examined to further investigate its regulatory functions in small intestinal development.

PDX1 is a homeodomain transcription factor and key regulator of genes that are involved in pancreatic development and intestinal differentiation [29,30] and, along with (sex-determining region Y)-box 2 and caudal type homeobox 2, it regulates the differentiation of enterocytes, Brunner’s gland cells, and enteroendocrine cells in the proximal intestine by modulating genes that are related to lipid metabolism and iron absorption [31,32,33]. We found that eckolinduced PDX1 activates *HBEGF* expression, which is implicated in wound healing, heart development, and gut protection [34,35,36]. HBEGF protects against intestinal hypoxia and ischemia/reperfusion injury, as well as necrotizing enterocolitis by suppressing pro-inflammatory cytokine-induced apoptosis and ROS production [37,38,39]. In addition to PDX1, HBEGF expression is regulated by nuclear factor-κB, specificity protein 1, and myogenic differentiation transcription factors in diverse biological processes, such as inflammation, homeostasis, and development [40,41,42]. In the present study, the PDX1 binding site (C/TTAATG) was detected in the upstream region (−954, −790, and −767) of the porcine *HBEGF* gene. Deletion analysis confirmed that PDX1 directly binds the sequence upstream of −790 to regulate basal transcriptional *HBEGF* activity in small intestinal epithelial cells.

The intestinal epithelium restricts paracellular penetration of potentially toxic substances that cause intestinal inflammation and injury through the tight and adherens junction complexes. It also supports wound repair by stimulating epithelial cell migration, proliferation, and differentiation [43,44,45]. HBEGF treatment was found to increase TEER and decrease FD-4 penetration that results from LPS-induced intestinal barrier dysfunction by increasing the expression of tight junction proteins, such as ZO-1 and occludin. Our results agree with those of earlier studies that reported that *HBEGF* knockout mice have shorter villi and higher mucosal permeability, as well as perturbed intestinal wound healing, which was reversed by HBEGF overexpression [46,47]. These findings suggest that HBEGF serves as a prophylactic or therapeutic agent for preventing or treating intestinal disorders, such as hypo-perfusion injury and peritonitis-induced sepsis. The extracellular signal-regulated kinase (ERK)1/2 MAPK, PI3K/AKT, nucleotide-binding oligomerization domain-containing protein 1/2, c-Jun N-terminal kinase (JNK)1/2, and STAT signaling pathways are involved in intestinal barrier function and wound healing [11,44,48]. The present study showed that HBEGF alleviated LPS-induced intestinal dysfunction through PI3K/AKT and P38 signaling pathway activation. In a previous report, HBEGF enhanced wound healing from intestinal ischemia/reperfusion injury through PI3K/AKT and MAPK/ERK kinase/ERK1/2 activation [49]. Our results also provide evidence that HBEGF preserves the barrier function of LPS-stimulated intestinal cells and promotes wound healing through P38 and PI3K/AKT signaling.

The present study confirmed that the eckol in EEEC reduces oxidative stress in intestinal epithelial cells. Oxidative stress that is caused by ROS or reactive nitrogen species is linked to various intestinal disorders, such as IBD, gastroduodenal ulcers, and colon cancer [50,51]. Polyphenols protects cells against oxidative stress by eliminating free radicals [52]. Eckol attenuates oxidative stress by upregulating transcription factors, including nuclear respiratory factor and Forkhead box O3a, which directly regulate ROS-scavenging enzymes, such as HO-1 and MnSOD through the NRF2/JNK, ERK, PI3K/AKT, and AMPK signaling pathways [25,53,54]. In the present study, we showed that eckol alleviates H_2_O_2_-induced oxidative stress by stimulating *HO-1* and *MnSOD* expression through the PI3K/AKT, P38, and AMPK signaling pathways, which is consistent with the findings of previous studies. Based on these results, we propose that eckol can be used as a therapeutic agent to alleviate oxidative stress.

We also investigated the in vivo effects of EEEC on growth, serum stress hormone, and oxidative stress levels in pigs during the weaning period. Dietary EEEC supplementation improved growth performance, and reduced the levels of stress hormones (cortisol, epinephrine, and norepinephrine) and antioxidants (SOD and GPx). The positive effect of eckol on growth might be due to its ability to improve intestinal dysfunction and reduce oxidative stress. The growth of piglets during the suckling-to-weaning transition is directly related to the total market days of the animal; for example, the number of days to market was approximately 6–10 days shorter for piglets with higher ADG (227 g/day) on the first week after weaning as compared with those with lower ADG (150 g/day) [1]. In the present study, EEEC as a feed additive supplement improved ADG at phases I (1–7 days) and II (7–21 days), but not at phase III (21–42 days). Based on this result, it is possible that EEEC was only effective during phase I and II development, since these phases are associated with rapid intestinal changes; however, it was not effective at phase III, because intestinal development has returned to normal. Thus, further studies are warranted in order to determine whether or not EEEC supplementation is directly related to a decrease in total market days in terms of pig production.

## 4. Materials and Methods

### 4.1. Animals and Feeding Trial

A total of 160 crossbred weaned pigs ([Yorkshire × Landrace] × Duroc) with an average body weight of 8.23 ± 0.93 kg were used in a six-week feeding trial. The pigs were sorted into pens (n = 5 per pen, eight pens per treatment) and into the following feeding groups: corn-soybean meal (C, control); control + 0.05% EEEC (T1); and, control + 0.1% EEEC (T2). Body weight was recorded at the beginning of the experiment (day 0) and on days 7, 21, and 42, and feed consumption was recorded for each pen in order to calculate the ADG and average daily feed intake in the experiment. Blood samples (10 mL) were collected from 10 random pigs per treatment group at the end of the feeding trial. Serum was separated through centrifugation at 4000× *g* for 30 min at 4 °C, and the aliquots were stored at 4 °C prior to determining the levels of stress hormones, such as cortisol, epinephrine, and norepinephrine; and, oxidative markers, such as SOD, MDA, and GPx.

Serum epinephrine, and norepinephrine levels were quantified through ion-exchange purification, followed by high-performance liquid chromatography with electrochemical detection. The serum SOD, MDA, and GPx levels were determined using enzyme-linked immunosorbent assay commercial kits (all from R&D Systems, Minneapolis, MN, USA).

### 4.2. Gene Expression Profiling

During the feeding trial, five piglets per treatment (control and T2) were sacrificed on day 14. The intestinal samples were pooled, frozen in liquid nitrogen, and stored at −80 °C. Total RNA was isolated using TRIzol reagent (Invitrogen, Carlsbad, CA, USA). Library construction was performed using the SENSE 3’ mRNA-Seq Library Prep Kit (Lexogen, Vienna, Austria), according to the manufacturer’s instructions. Moreover, an oligo-dT primer containing an Illumina-compatible sequence at the 5’ end was subjected to hybridization to 500 ng of the total RNA, followed by reverse transcription. After the degradation of the RNA template, second-strand synthesis was initiated with a random primer containing an Illumina-compatible linker sequence at the 5’ end. The double-stranded library was purified and amplified, and complete adapter sequences that were required for cluster generation were added. The finished library was purified from the PCR components. High-throughput sequencing was performed by single-end 75 sequencing on a NextSeq 500 instrument (Illumina, San Diego, CA, USA).

### 4.3. Quantitative Real-Time (qRT)-PCR and Western Blotting

For qRT-PCR, total RNA (1 µg) was used as a template for cDNA synthesis using the Maxima First-strand cDNA Synthesis Kit (Life Technologies, Carlsbad, CA, USA). The qRT-PCR primers for each target gene were designed while using the Primer3 program (http://frodo.wi.mit.edu/; Table 1), and the reaction was performed on a 7500 Fast Real-Time PCR System (Applied Biosystems, Foster City, CA, USA) under the following conditions: 94 °C for 3 min., followed by 40 cycles of 94 °C for 30 s, 59 °C–61 °C for 30 s, and 72 °C for 30 s. Target gene expression levels were normalized to that of the housekeeping gene *glyceraldehyde 3-phosphate dehydrogenase* and calculated using the 2^−ΔΔCT^ method.

For Western blotting, cultured IPEC-J2 cells that were subjected to various treatments were lysed using a lysis buffer (Cell Signaling Technology, Danvers, MA, USA) containing a protease inhibitor cocktail (Roche, Basel, Switzerland). The protein concentration was determined using a Pierce BCA Protein Assay Kit (Thermo Fisher Scientific, Waltham, MA, USA), and the proteins in each sample (20 μg) were separated through electrophoresis on a 10% polyacrylamide gel for 1 h at 120 V and transferred to a nitrocellulose membrane (Millipore, Billerica, MA, USA) while using a Mini-PROTEAN Tetra Cell (Bio-Rad, Hercules, CA, USA). After blocking for 1 h, the membrane was incubated overnight at 4 °C with the appropriate primary antibodies, including anti-HBEGF and anti-PDX1 (both from Antibodies-online GmbH, Aachen, Germany), anti-AKT and anti-P38 (both from Cell Signaling Technology, USA), and anti-ZO-1 (Thermo Fisher Scientific, Waltham MA, USA). After three washes with Tris-buffered saline containing 0.1% Tween 20, the membrane was incubated for 1 h at room temperature with the appropriate secondary antibodies. Immunoreactivity was visualized with ECL Select Western blotting detection reagent (GE Healthcare, Little Chalfont, UK), and the protein bands were imaged while using the ChemiDoc imaging system (Bio-Rad, Hercules, CA, USA). Densitometric analysis was performed using the ImageJ software (National Institutes of Health, Bethesda, MD, USA).

### 4.4. Cell Culture and Treatments

The IPEC-J2 intestinal porcine enterocyte cell line (DSMZ, Braunschweig, Germany) was maintained, as previously described [11]. Moreover, the cells were cultured in high-glucose Dulbecco’s Modified Eagle’s Medium supplemented with 5% fetal bovine serum, 1% insulin–transferrin–selenium-X, and 1% (*v/v*) penicillin–streptomycin [55]. The cells were maintained at 37 °C in a humidified atmosphere of 5% CO_2_. The IPEC-J2 cells were incubated with different treatments, including recombinant HBEGF (Mybiosource, San Diego, CA, USA), LY2944002 (Cell Signaling Technology, Danvers, MA, USA), SB202190 (Cell Signaling Technology, USA), and the AMPK inhibitor (dorsomorphin dihydrochloride; Santa Cruz Biotechnology, Dallas, TX, USA) at different intervals, as indicated in the figure legends.

### 4.5. Immunofluorescence

IPEC-J2 cell monolayers on glass coverslips were fixed with 4% paraformaldehyde. After blocking with 2% bovine serum albumin in phosphate-buffered saline, the cells were incubated overnight at 4 °C with primary antibodies against zona occludens (ZO)-1 and occludin diluted to a 1:100 ratio. After washing, fluorophore-conjugated secondary antibody (Alexa Fluor 488) was applied for 1 h at room temperature in the dark. The coverslips were mounted on slides while using Vectashield Antifade Mounting Medium with DAPI (Vector Laboratories, Burlingame, CA, USA), and the images were obtained using a fluorescence microscope.

### 4.6. Cell Proliferation and Migration Assays

To evaluate proliferation, the IPEC-J2 cells were seeded in 96-well plates at a density of 1 × 10^4^ cells/well. Water-soluble tetrazolium-1 cell proliferation reagent (Roche Applied Science, Indianapolis, IN, USA) was added to each well, and dye absorbance at the end of the incubation period was measured at 450 nm with background subtraction at 690 nm while using a BioTek Synergy HTTR microplate reader (BioTek Instruments, Winooski, VT, USA).

For the migration assay, the IPEC-J2 cells were cultured with various compounds (eckol, heparin-binding epidermal growth factor-like growth factor (HBEGF), and inhibitors) in 60-mm culture dishes until reaching confluence. A straight scratch was made using a P200 pipette tip across the bottom of the dish, and the migrating cells were photographed at different times.

### 4.7. Transepithelial Electrical Resistance (TEER) and Intestinal Permeability

Confluent IPEC-J2 cell monolayers were cultured in 24-well transwell chambers (polycarbonate membrane, filter pore size = 0.4 μm, area = 0.33 cm^2^; Costar) under different treatment conditions for 24 h. The cells were washed twice and incubated with lipopolysaccharide (LPS; 1 μg/μL) for 1 h. TEER was measured using an epithelial voltohmmeter (World Precision Instruments, Sarasota, FL, USA). TEER values were calculated by subtracting the blank filter (90 Ω) and multiplying by the surface area of the filter. All of the measurements were performed for at least three wells.

The IPEC-J2 cells growing in a confluent monolayer (≥1 kΩ cm^2^) were treated with different compounds for 24 h. The cells were washed twice and then incubated with LPS for 1 h. After two additional washes, the permeability assay was initiated by adding 500 μL of culture media containing 1 mg/mL fluorescein isothiocyanate–dextran (FD)-4 (Sigma–Aldrich, St. Louis, MO, USA) to the apical chamber, whereas the basolateral chamber was filled with 1.5 mL of culture media. FD-4 was allowed to permeate overnight (18 h at 37 °C and 5% CO_2_) from the apical to the basolateral chamber, and 100 μL of the basolateral chamber media was transferred to a 96-well plate in order to measure the extent of FD-4 permeation while using a fluorometer (excitation, 490 nm; emission, 520 nm).

### 4.8. Cellular ROS Detection

The IPEC-J2 cells were seeded in a clear-bottomed 96-well black plate at a density of 1 × 10^4^ cells/well. After eckol and inhibitor treatment with or without H_2_O_2_, the cells were stained with 2.5 μM 2’,7’-dichlorofluorescein (DCF) diacetate, and fluorescence intensity was measured using a fluorometer at excitation and emission wavelengths of 485 and 535 nm, respectively. The values are expressed as fold increase relative to control cell values.

### 4.9. Vector construction, Gene Silencing, and Luciferase assay

For *pancreatic and duodenal homeobox* (*PDX*)*1* knockdown, the IPEC-J2 cells were transfected with specific small interfering (si)RNAs (Appendix A) while using RNAiMAX (Invitrogen), according to the manufacturer’s instructions. After 24 h, total RNA was extracted and analyzed using qRT-PCR.

For the promoter assay, upstream sequences of the *HBEGF* gene and the −1000 upstream sequence containing deletions of the *PDX1*-binding site synthesized by Bioneer (Daejeon, Korea) were sub-cloned into the pGL3-Basic vector. Each vector contained a different length of the *HBEGF* upstream sequence, and the vector was transfected into the IPEC-J2 cells. After 4 h, the cells were analyzed using a luciferase assay kit (Promega, Madison, WI, USA). Moreover, the cells were re-suspended in 100 μL of diluted Passive Lysis Buffer, and the extracts were centrifuged at 13,000× *g* for 5 min. Firefly and *Renilla* luciferase activities in the supernatant were quantified while using a GLOMAX 20/20 luminometer (Promega). The measured values were normalized to *Renilla* activity.

### 4.10. Statistical Analysis

Data were analyzed by evaluating the differences among treatments using Duncan’s multiple range tests through the general linear model function of the SAS software (Systat, Cary, NC, USA). The results are expressed as the mean ± standard error (n ≥ 3, where n is the number of replicates). A *p* value of <0.05 was considered to be statistically significant.

## 5. Conclusions

The present study demonstrated that eckol enhanced intestinal function in pigs during the suckling-to-weaning transition by improving intestinal barrier function and wound healing, as well as reducing oxidative stress through PDX1-induced HBEGF expression (Figure 9). Our results indicated that eckol as a feed additive can improve the overall health and growth of livestock by preventing intestinal dysfunction during this critical developmental period, which can subsequently increase livestock marketability.

## Figures and Tables

**Figure 1 ijms-21-04755-f001:**
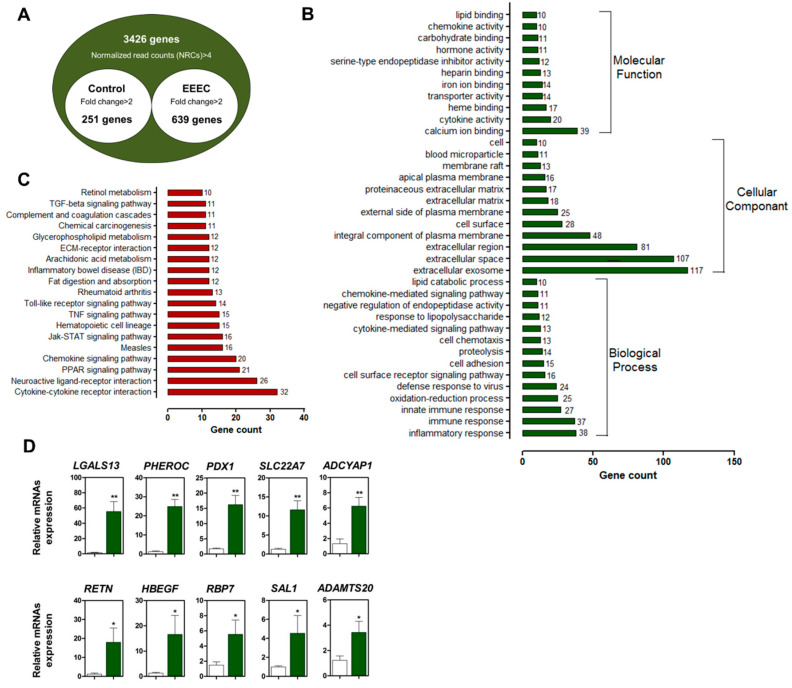
Gene expression profiling of the small intestine of pigs treated with the ethanol extract of dried *E. cava* (EEEC). (**A**) Venn diagram of genes up- or downregulated at least two-folds after EEEC treatment, compared with the control levels. (**B**) GO terms assigned to “biological processes”, “cellular components”, and “molecular functions” (*p* < 0.01). (**C**) Kyoto Encyclopedia of Genes and Genomes (KEGG) pathway analysis of the same gene sets (*p* < 0.01). (**D**) Quantitative analysis of the top 10 differentially expressed genes (DEGs; n = 3). Relative expression levels were normalized to those of *glyceraldehyde 3-phosphate dehydrogenase* (*GAPDH*). Error bars indicate standard error of triplicate analyses. * *p* < 0.05, ** *p* < 0.01.

**Figure 2 ijms-21-04755-f002:**
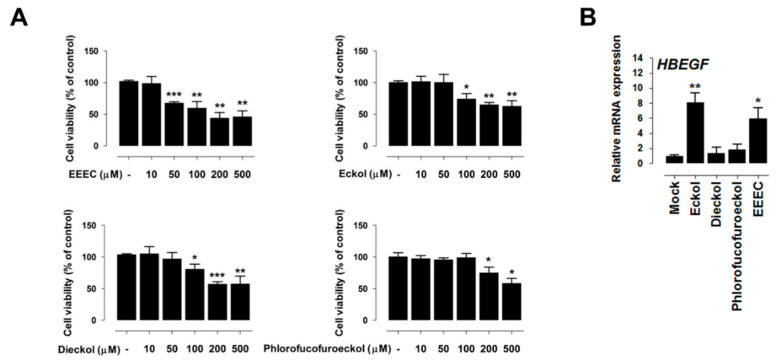
Toxicity of ethanol extracted of dried *E. cava* (EEEC), eckol, dieckol, and phlorofucofuroeckol (**A**) EEEC, eckol, dieckol, and phlorofucofuroeckol toxicity was evaluated by performing a cell viability assay using IPEC-J2 cells incubated with the indicated concentrations of each compound for 24 h. (**B**) Heparin-binding epidermal growth factor-like growth factor (*HBEGF)* expression in the treatment groups (EEEC, eckol, dieckol, and phlorofucofuroeckol). In qRT-PCR analysis, target gene expression levels were normalized to those of *glyceraldehyde 3-phosphate dehydrogenase* (*GAPDH*). Error bars indicate standard error of triplicate analyses. * *p* < 0.05, ** *p* < 0.01, *** *p* < 0.001.

**Figure 3 ijms-21-04755-f003:**
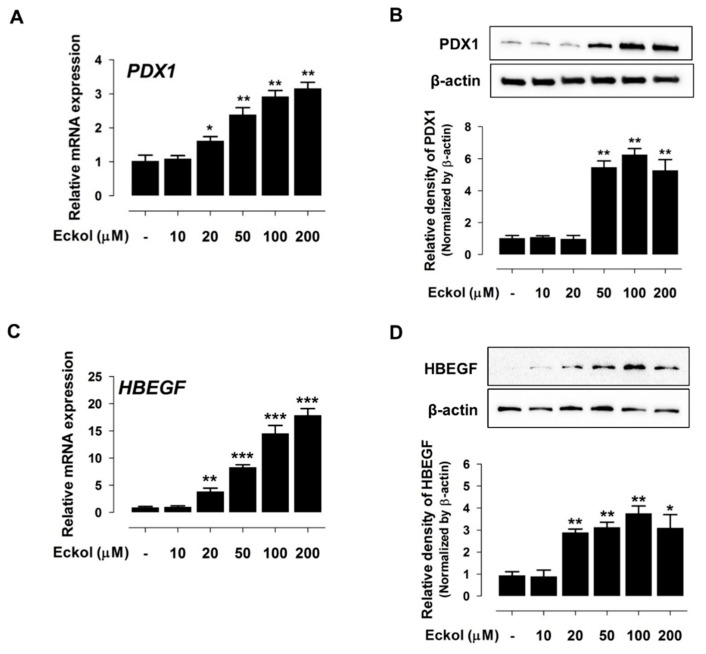
Eckol induces PDX1 and HBEGF expression. PDX1 mRNA (**A**) and protein (**B**) expression in cells treated with eckol at different concentrations (0 µM, 10 µM, 20 µM, 50 µM, 100 µM, and 200 µM). HBEGF mRNA (**C**) and protein (**D**) expression in cells treated with various concentrations of eckol. Protein band intensity in the Western blot analysis was quantified using the threshold function of the ImageJ software. For qRT-PCR analysis, target gene expression levels were normalized to those of *glyceraldehyde 3-phosphate dehydrogenase* (*GAPDH*). Error bars indicate standard error of triplicate analyses. * *p* < 0.05, ** *p* < 0.01, *** *p* < 0.001.

**Figure 4 ijms-21-04755-f004:**
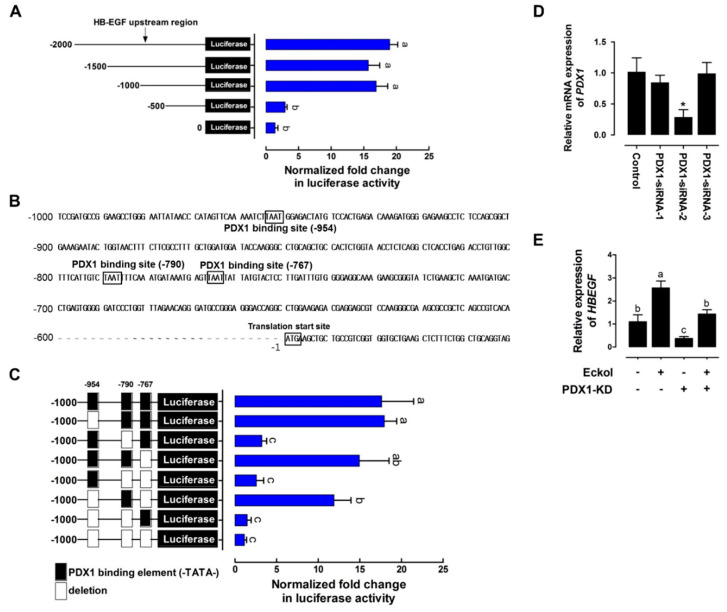
Eckol-induced PDX1 regulates HBEGF expression. (**A**) Activity of *HBEGF* upstream region. IPEC-J2 cells were transfected with DNA sequences of different lengths (−2000, −1500, −1000, and −500) comprising the upstream region of *HBEGF* (n = 3). (**B**) Upstream sequence of the putative core region of the *HBEGF* promoter. Sequence numbering is relative to the transcription start site. Putative PDX1 binding sites (TAAT) are boxed and labeled above (upstream of −954, −790, and −767). (**C**) Deletion assay of putative PDX1 binding sites. Sequences in which the deleted binding sites (upstream of −954, −790, and −767) were transfected into IPEC-J2 cells (n = 3). (**D**) *PDX1* knockdown assay. The siRNA-mediated suppression of *PDX1* in IPEC-J2 cells was confirmed using qRT-PCR. (**E**) Relative *HBEGF* expression in *PDX1*-silenced IPEC-J2 cells treated with eckol. Relative luciferase activity was calculated as the ratio of firefly to *Renilla* luciferase, and the relative expression level was normalized to that of *glyceraldehyde 3-phosphate dehydrogenase* (*GAPDH*). Error bars indicate standard error of triplicate analyses. * *p* < 0.05. Lower-case letters indicate significant differences among treatments based on Duncan’s multiple range test.

**Figure 5 ijms-21-04755-f005:**
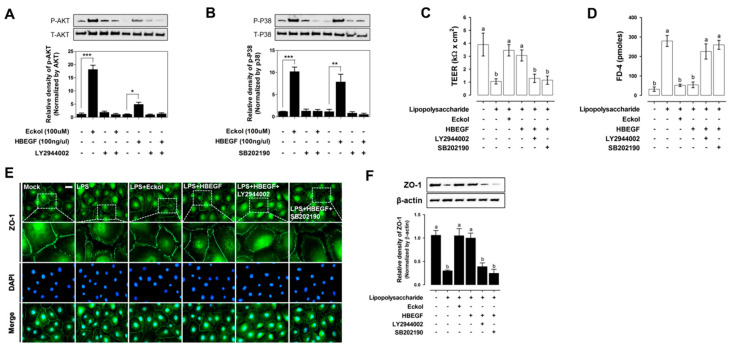
Effects of eckol on intestinal barrier function. Activation of PI3K/AKT (**A**) and P38 (**B**) signaling pathways in IPEC-J2 cells treated with eckol and HBEGF for 2 h under lipopolysaccharide (LPS) challenge. AKT (Ser473) and P38 (Thr180/Tyr182) phosphorylation was evaluated by Western blotting. Protective effects of eckol and HBEGF on intestinal barrier function in IPEC-J2 cells under LPS challenge, as determined by TEER quantification (**C**), and the permeability of fluorescein isothiocyanate-FD-4 (**D**) (n = 3). ZO-1 expression upon eckol and HBEGF treatment under LPS challenge was evaluated by immunofluorescence analysis (**E**) and Western blotting (**F**). Nuclei were stained with 4’,6-diamidino-2-phenylindole (DAPI). IPEC-J2 cells were treated with eckol and HBEGF under LPS challenge with or without PI3K/AKT (LY2944002) and P38 (SB202190) inhibitors. Error bars indicate standard error of triplicate analyses. * *p* < 0.05, ** *p* < 0.01, *** *p* < 0.001. Lower-case letters indicate significant differences among treatments that are based on Duncan’s multiple range test.

**Figure 6 ijms-21-04755-f006:**
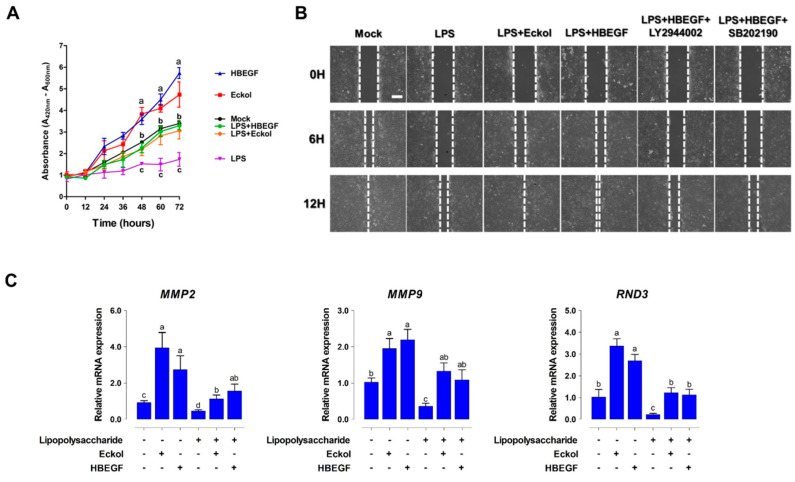
Eckol promotes intestinal wound healing through HBEGF. (**A**) Cell proliferation was assessed at 12 h, 24 h, 36 h, 48 h, 60 h, and 72 h using the water-soluble tetrazolium-1 assay (n = 3) (**B**) Migration was assessed at 0 h, 6 h, and 12 h using a wound-healing assay (n = 3) (**C**) Relative expression levels of cell migration-related genes were analyzed using qRT-PCR in eckol- and HBEGF-treated IPEC-J2 cells under LPS challenge with or without PI3K/AKT (LY2944002) and P38 (SB202190) inhibitors. Error bars indicate standard error of triplicate analyses. Lower-case letters indicate significant differences (*p* < 0.05) among treatments, based on Duncan’s multiple range test.

**Figure 7 ijms-21-04755-f007:**
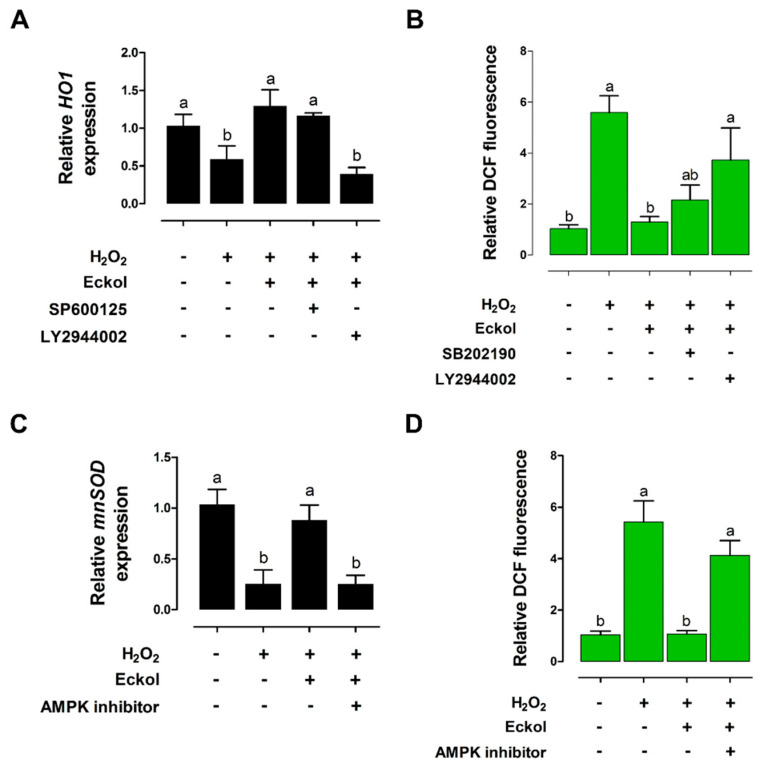
Eckol protects against H_2_O_2_-induced oxidative stress in IPEC-J2 cells. Relative *HO-1* (**A**) and *MnSOD* expression levels (**C**) were evaluated through qRT-PCR (n = 3). Relative cellular ROS ratio was determined through 2’,7’-dichlorofluorescein (DCF) fluorescence analysis with or without PI3K/AKT (LY2944002), P38 (SB202190) (**B**), or 5’-AMP-activated protein kinase (AMPK) (**D**) inhibitor (n = 3) in eckol-treated cells exposed to H_2_O_2_-induced oxidative stress. Lower-case letters indicate significant differences (*p* < 0.05) among treatments, based on Duncan’s multiple range test.

**Figure 8 ijms-21-04755-f008:**
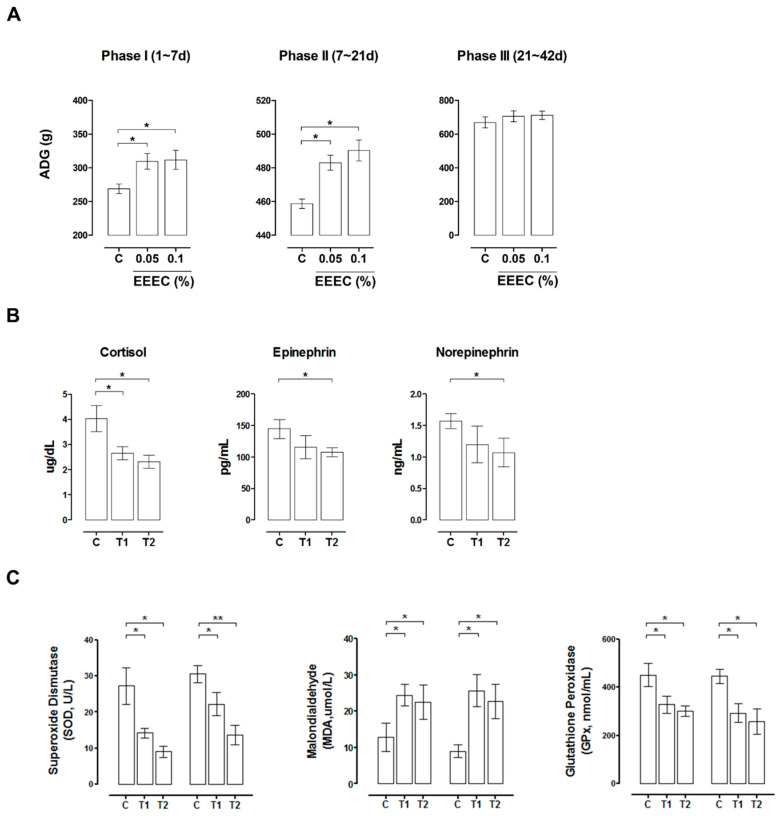
In vivo feeding trial during the suckling-to-weaning transition in pigs. The pigs were sorted into pens (n = 5 per pen, eight pens per treatment) and assigned to the following feeding groups: corn-soybean meal (C, control); control + 0.05% EEEC (T1); control + 0.1% EEEC (T2). (**A**) Dietary ethanol extract of dried *E. cava* (EEEC) supplementation improved average daily gain (ADG) in pigs during three developmental phases (days 1–7, 7–21, and 21–42). Serum levels of stress-related hormones, such as cortisol, epinephrine, and norepinephrine (**B**); and oxidative stress markers, such as superoxide dismutase (SOD), malondialdehyde (MDA), and glutathione peroxidase (GPx) (**C**), were determined through an enzyme-linked immunosorbent assay (n = 10). Error bars indicate the standard error of triplicate analyses. * *p* < 0.05 and ** *p* < 0.01

**Figure 9 ijms-21-04755-f009:**
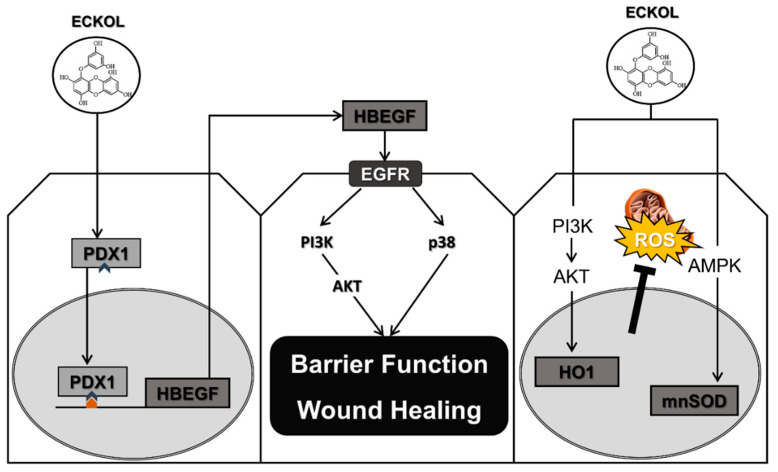
Schematic illustration of the effects of eckol on intestinal development during the suckling-to-weaning transition. Eckol induces PDX1 expression that directly regulates *HBEGF* expression, leading to enhanced intestinal barrier function, accelerated wound healing, and reduced oxidative stress through the activation of various intracellular signaling pathways.

**Table 1 ijms-21-04755-t001:** List of primers.

Gene Symbol	Description	Accession No.	Primer Sequence (5’−>3’)	
			Forward	Reverse
*LGALS13*	lectin, galactoside-binding, soluble, 13	NM_001142841	CTCTCGCCACAATCTGTGAA	ATCCCGTTTGTGAACTCAGG
*PHEROC*	pheromaxein C subunit	NM_001123161	CCAGTGATTCCAGCGTAACC	CACTTGCATAAACACGCTGAA
*PDX1*	pancreatic and duodenal homeobox 1	NM_001141984	TGAAATTGATGCTGGTGGAA	CATGGGGAGTACAGGCACTT
*SLC22A7*	solute carrier family 22 member 7	NM_001044617	GGGAAGGGTTTTTCTGAAGC	TGACAGCCATACTCCATCCA
*ADCYAP1*	adenylate cyclase activating polypeptide 1	NM_001001544	ACAGCAGCGTCTACTGCTCA	TCTCTTTCTTCCGCTGGGTA
*RETN*	resistin	NM_213783	CTCAGGCTTTGCTGTCACTG	GATGCGCAGATGCAAACTTA
*HBEGF*	heparin-binding EGF like growth factor	NM_214299	CCTACCGAATCTACGGACCA	CTTTCTTTTCCCTCGCTCCT
*RBP7*	retinol binding protein 7	NM_001145222	TGCTGGCCCTAGGTATTGAC	CCAGGCCTTTGTTATCCTCA
*SAL1*	salivary lipocalin	NM_213814	ATGTCAATGGCGACAAAACA	AGTTGGAAGCAGCGATCAAT
*ADAMTS20*	ADAM metallopeptidase with thrombospondin type 1 motif 20	NM_001257275	CATCAGCTGTGGCCTGTAGA	CGGCCATACATTCCACTCTT
*HO1*	HMOX1, heme oxygenase 1	NM_001004027	AGCTGTTTCTGAGCCTCCAA	GAACGAAGAGTGGCTCCAAC
*mnSOD*	SOD2, superoxide dismutase 2	NM_214127	TTTGGGGCTGTTTTTGTAGG	TGATGGTTTGGGATGGTTTT
*MMP2*	matrix metallopeptidase 2	NM_214192	ACTCCCACTTTGACGACGAT	CGTACTTGCCATCCTTGTCG
*MMP9*	matrix metallopeptidase 9	NM_001038004	TGAAGACGCAGAAGGTGGAT	TTCAGGAGGTCGAAGGTCAC
*CDH1*	cadherin 1	NM_001163060	CTGTTGCAGGTCTCATCGTG	AACATAGACCGTCCTTGGCA
*RND3*	Rho family GTPase 3	NM_214296	GAGAGAAGAGCCAGCCAGAA	TGTCCCACAGGCTCAATTCT
*GAPDH*	Glyceraldehyde-3-phosphate dehydrogenase	NM_001206359	ACACCGAGCATCTCCTGACT	GACGAGGCAGGTCTCCCTAA

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
