# Peer review of "Eckol Alleviates Intestinal Dysfunction during Suckling-to-Weaning Transition via Modulation of PDX1 and HBEGF"

_ijms, 2020, doi:10.3390/ijms21134755_

Round 1
Reviewer 1 Report
The publication submitted for review concerns the gene expression and intestinal function after treatment with eckol during the suckling-to-weaning transition. Obtained results can not only improve animal health but also increase economic productivity. The experience is well planned, uses many research methods and leads to a consistent conclusion. However, some doubts required authors’ comments:
- In a feeding trial authors used 3 groups of animals (C, T1 and T2)- why in gene expression profiling they used only 2 groups?
- Why animals were sacrificed on day 14?
Nevertheless, these comments are not critical to the quality of the work, which I consider high and I support the publication of the manuscript in the IJMS.
Author Response
Cover letter
To the Editor
Please find enclosed the revised manuscript (ijms-844841) entitled “Eckol alleviates intestinal dysfunction during suckling-to-weaning transition via modulation of PDX1 and HBEGF” by Lee et al. I would like to express my sincere gratitude to the editor and reviewers for their valuable suggestions and comments, which have greatly helped us to improve our manuscript. This manuscript has been substantially revised to improve the strength and impact of our work. We have enclosed a point-by-point response to the reviewer’s comments. Please feel free to contact me if you need further clarification.
Thank you for your consideration.
Response to reviewer
# Reviewer 1
The publication submitted for review concerns the gene expression and intestinal function after treatment with eckol during the suckling-to-weaning transition. Obtained results can not only improve animal health but also increase economic productivity. The experience is well planned, uses many research methods and leads to a consistent conclusion. However, some doubts required authors’ comments:
- In a feeding trial authors used 3 groups of animals (C, T1 and T2)- why in gene expression profiling they used only 2 groups?
Response:
We appreciate the reviewer’s helpful comment. As mentioned, the present study profiled gene expressions from the control and T2 (control + 0.1% EEEC) groups. Unfortunately, we have not done gene expression profiling for the T1 (control + 0.1% EEEC) group; however, we surmised that gene expression profiling of the control and T2 groups is enough to identify differentially expressed genes that were affected by EEEC treatment, based on the results of the feeding trial (Figure 8), which shows that the effects of EEEC supplementation on growth performance, stress hormone activity, and oxidative stress levels were in a concentration-dependent manner.
- Why animals were sacrificed on day 14?
Thank you for raising an important question. As mentioned in the Introduction section, the intestine undergoes marked structural and functional changes, including villous atrophy and crypt elongation, which decrease its absorptive capacity, thereby influencing feeding efficiency during suckling-to-weaning transition. In many previous reports, physiologic changes of the intestine, including structure and digestive enzyme activities, continued until approximately 15 days after weaning. Thus, we performed gene expression profiling at day 14 after weaning.
Pluske JR, Hampson DJ, Williams IH: Factors influencing the structure and function of the small intestine in the weaned pigs: a review. Livest Prod Sci 1997, 51:215–236.
Lalles J, Boudry G, Favier C, LeFloc N, Luron I, Montagne L, Oswald IP, Pié S, Piel C, Sève B: Gut function and dysfunction in young pigs: physiology. Anim Res 2004, 53:301–316.
- Nevertheless, these comments are not critical to the quality of the work, which I consider high and I support the publication of the manuscript in the IJMS.
Response:
We highly appreciate the reviewer’s constructive comments.

Reviewer 2 Report
ijms-844841
In this manuscript, Lee and Kim investigate the ability of a polyphenol (Eckol) extracted from E. cava to alleviate intestinal stress during suckling-to-weaning transition in pigs. Whereas the integrity of intestinal barrier appears to be mediated by HBEGF induced by Eckol only in presence of functional PDX1, the protection induced by EEEC from oxidative stress seems to be independent form HBEGF activity. Unfortunately, the manuscript does not provide evidence for in vivo significance of Eckol usage alone.
The paper is well written and explores a very peculiar physiological transition, but some concerns remains.
MAJOR POINTS
- Line 134. The sentence sound like a statement. If this is the case, please indicate a reference. Otherwise, rewrite the sentence.
- Figure 5 and Figure 6. Authors should investigate Eckol treatment with silenced-HBEGF expression in presence of LPS. Does Eckol have still the same effect?
- Figure 5 and Figure 6. Combined treatment with Eckol and HBGEF is missing. Does this combination have a synergic effect?
- Figure 6. What are the expression levels of other genes involved in migration (CDX2, PCNA, CDH1)?
- Figure 7. What are the levels of HO1 and mnSOD with Eckol alone?
- Figure 8A. In the phase I, for T1 and T2, it looks like there are outliers. A box plot with dots maybe represent the best way to depict this data. Moreover, legend indicate that all the 3 phases are improved by Eckol treatment. Please, rewrite the legend according to the data.
- EEEC treatment alleviates stressor. From the manuscript, it seems that this is independent form HBEGF. What are the levels of HBEGF in both T1 and T2?
- Is Eckol alone able to alleviates stress in vivo? Or only EEEC does?
- It would be interesting evaluate the permeability to nutrients (glucose/protein/fatty acids) in the intestine during suckling to weaning transition both in normal condition and after Eckol administration.
- Lines 277-280. Eckol is able to induce PDX and HBEGF pathway but we do not know if it is the main active compound.
MINOR POINTS:
- Please indicate the sequence of primers used.
Author Response
Cover letter
To the Editor
Please find enclosed the revised manuscript (ijms-844841) entitled “Eckol alleviates intestinal dysfunction during suckling-to-weaning transition via modulation of PDX1 and HBEGF” by Lee et al. I would like to express my sincere gratitude to the editor and reviewers for their valuable suggestions and comments, which have greatly helped us to improve our manuscript. This manuscript has been substantially revised to improve the strength and impact of our work. We have enclosed a point-by-point response to the reviewer’s comments. Please feel free to contact me if you need further clarification.
Thank you for your consideration.
Response to reviewer
# Reviewer 2
In this manuscript, Lee and Kim investigate the ability of a polyphenol (Eckol) extracted from E. cava to alleviate intestinal stress during suckling-to-weaning transition in pigs. Whereas the integrity of intestinal barrier appears to be mediated by HBEGF induced by Eckol only in presence of functional PDX1, the protection induced by EEEC from oxidative stress seems to be independent form HBEGF activity. Unfortunately, the manuscript does not provide evidence for in vivo significance of Eckol usage alone.
The paper is well written and explores a very peculiar physiological transition, but some concerns remains.
MAJOR POINTS
- Line 134. The sentence sound like a statement. If this is the case, please indicate a reference. Otherwise, rewrite the sentence.
Response:
According to reviewer’s suggestion, we have modified sentence in line 144-147 of the revised manuscript, as follows: “To investigate whether PDX1 activation directly regulates HBEGF expression in intestinal epithelial cells in response to eckol treatment, we cloned different lengths of porcine HBEGF gene promoter sequences into the firefly luciferase plasmid to evaluate the transcriptional activity of PDX1 in response to eckol treatment”.
- Figure 5 and Figure 6. Authors should investigate Eckol treatment with silenced-HBEGF expression in presence of LPS. Does Eckol have still the same effect?
Response:
We appreciate the reviewer’s suggestion that the authors should investigate eckol treatment with silenced-HBEGF expression in the presence of LPS to examine whether eckol still has the same effect or not. The purpose of the present study is to investigate effects of eckol on intestinal barrier function and wound healing through different signaling pathways, such as phosphoinositide 3-kinase (PI3K)/AKT and P38 signaling pathways through PDX1-mediated HBEGF modulation. Thus, we focused on the whether eckol induced PDX1 transcription factor expression and whether PDX1 directly regulates HBEGF expression by binding to the promoter region of HBEGF. As per the reviewer’s suggestion, we agree that it would be more accurate to investigate the function of eckol when HBEGF expression was suppressed. In fact, experiments on the relationship between the effects of eckol and suppressed HBEGF expression were considered in the design of the experiment, but the authors decided to substitute the results for related signals, including inhibitor treatment, instead of suppressing HBEGF expression. If the reviewer would require such excluded experiment, we will be glad to add experimental data, but we would humbly ask that the reviewer allow us approximately 1–2 months to do so.
- Figure 5 and Figure 6. Combined treatment with Eckol and HBGEF is missing. Does this combination have a synergic effect?
Response:
We appreciate the reviewer’s comment regarding the authors’ need to examine combined eckol and HBGEF treatment in order to investigate their synergic effect. As per reviewer’s suggestion, we agree that investigating this possible synergic effect can enrich our results. However, the present study investigated the separate effects of eckol and HBEGF treatments on intestinal barrier function and wound healing through various signaling pathways, such as phosphoinositide 3-kinase (PI3K)/AKT and P38 signaling through PDX1-mediated HBEGF modulation. If the reviewer's judgment would require an experiment to look into the synergistic effect of eckol and HBEGF, it will be possible to add experimental data by giving the authors a time of approximately 1–2 months.
- Figure 6. What are the expression levels of other genes involved in migration (CDX2, PCNA, CDH1)?
Response:
Thank you for raising an important question. Unfortunately, we have not done an analysis on the expression of CDX2 and PCNA as wound-healing-related genes. However, we analyzed CDH1 expression pattern, which is also a wound-healing-related gene.
We could not accurately interpret the results and find meaning, because in contrast to the present study, several previous studies demonstrated that LPS decreased CDH1 expression. Thus, we did not include the result of the CDH1 expression pattern analysis in the submitted manuscript.
- Figure 7. What are the levels of HO1 and mnSOD with Eckol alone?
Response:
Thank you for clarifying this important point. Unfortunately, we have not done an analysis on the HO1 and mnSOD expression following eckol treatment in Figure 7. The present study focused on the effects of eckol under oxidative stress. As per the reviewer’s suggestion, we agree that investigating HO1 and mnSOD expression following eckol treatment can enrich our results. If the reviewer's judgment requires such an experiment, it will be possible to add experimental data by giving the authors an additional month to do so.
- Figure 8A. In the phase I, for T1 and T2, it looks like there are outliers. A box plot with dots maybe represent the best way to depict this data. Moreover, legend indicate that all the 3 phases are improved by Eckol treatment. Please, rewrite the legend according to the data.
Response:
According to the reviewer’s suggestion, we have modified Figure 8A (bar grape) in the revised manuscript.
- EEEC treatment alleviates stressor. From the manuscript, it seems that this is independent form HBEGF. What are the levels of HBEGF in both T1 and T2?
Response:
The authors performed gene expression profiling on the control and T2 (control + 0.1% EEEC) groups. Unfortunately, we have not done gene expression profiling on the T1 (control + 0.1% EEEC) group. However, we surmise that the data of gene expression profiling from the control and T2 groups were enough to identify the differentially expressed genes that were affected by EEEC treatment. Also, in Figure 1D, we verified the expression of the top 10 DEGs in the small intestine with or without 0.1% EEEC treatment through qRT-PCR, and it was confirmed that HBEGF was upregulated in the EEEC treatment group relative to its expression in the untreated group.
- Is Eckol alone able to alleviate stress in vivo? Or only EEEC does?
Response:
Thank you for raising this important point. The authors investigated the in vivo effects of EEEC as a feed additive on growth, serum stress hormone, and oxidative stress levels in pigs during the weaning period. Dietary EEEC supplementation improved growth performance, and reduced the levels of stress hormones (cortisol, epinephrine, and norepinephrine) and antioxidants (SOD and GPx). Unfortunately, we have not done a feeding trial to examine effects of eckol due to price and quantitative limitation. However, we inferred that the positive effect of eckol on growth may be due to its ability to improve intestinal dysfunction and reduce oxidative stress, based on the EEEC feeding trial.
- It would be interesting evaluate the permeability to nutrients (glucose/protein/fatty acids) in the intestine during suckling to weaning transition both in normal condition and after Eckol administration.
Response:
Thank you for providing these insights. Unfortunately, the present study did not evaluate the difference in the intestine’s permeability to nutrients during the suckling-to-weaning transition between with or without eckol administration. However, in future studies, we will overcome several limitations, such as the fluorescence labeling of nutrients and treatment methods of labeling nutrients in vivo, and we will try to perform in vivo permeability experiments both under normal conditions and after eckol administration.
- Lines 277-280. Eckol is able to induce PDX and HBEGF pathway but we do not know if it is the main active compound.
Response:
We appreciate the reviewer’s comment regarding eckol’s ability to induce PDX- and HBEGF-related pathways, but we do not know if it is the main active compound. Thus, we have modified the sentence in line 306-309 of the revised manuscript as follows: “Based on the results, eckol induced PDX1 and HBEGF expression to a greater extent than dieckol and phlorofucofuroeckol; hence, eckol was examined to further investigate its regulatory functions in small intestinal development”.
MINOR POINTS:
- Please indicate the sequence of primers used.
Response:
We appreciate the reviewer’s suggestion to indicate the sequence of primers in the main text. Previously, the primer list was indicated in a Supplemental Table of the original manuscript. According to the reviewer’s suggestion, we indicated the list of primers in the main text of the revised manuscript.

Round 2
Reviewer 2 Report
ijms-844841- R1
Authors replied to most of the comments. Other considerations remained without an answer mostly due to shortage of time. Anyway, I believe that the work is valuable and it is fine how it is.